# New Series of Zaxinone Mimics (MiZax) for Fundamental and Applied Research

**DOI:** 10.3390/biom13081206

**Published:** 2023-08-01

**Authors:** Muhammad Jamil, Pei-Yu Lin, Lamis Berqdar, Jian You Wang, Ikuo Takahashi, Tsuyoshi Ota, Noor Alhammad, Guan-Ting Erica Chen, Tadao Asami, Salim Al-Babili

**Affiliations:** 1The BioActives Lab, Center for Desert Agriculture, King Abdullah University of Science and Technology (KAUST), Thuwal 23955-6900, Saudi Arabia; muhammad.jamil@kaust.edu.sa (M.J.); peiyu.lin@kaust.edu.sa (P.-Y.L.); lamis.berqdar@kaust.edu.sa (L.B.); jianyou.wang@kaust.edu.sa (J.Y.W.); ny.alhammad@gmail.com (N.A.); guanting.chen@kaust.edu.sa (G.-T.E.C.); 2Plant Science Program, Biological and Environmental Science and Engineering Division, King Abdullah University of Science and Technology (KAUST), Thuwal 23955-6900, Saudi Arabia; 3Applied Biological Chemistry, The University of Tokyo, Tokyo 113-8657, Japan; takahashi.190@gmail.com (I.T.); higenomoto.co@gmail.com (T.O.); asami@mail.ecc.u-tokyo.ac.jp (T.A.)

**Keywords:** zaxinone, mimics of zaxinone (MiZax), plant growth regulator, strigolactones, *Striga hermonthica*

## Abstract

The apocarotenoid zaxinone is a recently discovered regulatory metabolite required for proper rice growth and development. In addition, zaxinone and its two mimics (MiZax3 and MiZax5) were shown to have a remarkable growth-promoting activity on crops and a capability to reduce infestation by the root parasitic plant *Striga* through decreasing strigolactone (SL) production, suggesting their potential for application in agriculture and horticulture. In the present study, we developed a new series of MiZax via structural modification of the two potent zaxinone mimics (MiZax3 and MiZax5) and evaluated their effect on plant growth and *Striga* infestation. In general, the structural modifications to MiZax3 and MiZax5 did not additionally improve their overall performance but caused an increase in certain activities. In conclusion, MiZax5 and especially MiZax3 remain the likely most efficient zaxinone mimics for controlling *Striga* infestation.

## 1. Introduction

The pressure on food demand is increasing due to a growing world population, increasing the risk of starvation and malnutrition [1]. Indeed, according to the United Nations Food and Agriculture Organization (FAO), our food production must be nearly tripled by 2050 to meet the needs of the expanding human population. Particularly, countries in sub-Saharan Africa are at high risk of food insecurity due to low agricultural production, caused by socio-economic conditions, and unfavorable biotic and abiotic factors, such as poor soil fertility, drought, pest attack, and weeds [2,3,4]. Indeed, weed infestation, especially of parasitic weeds, is one of the main reasons for low yield in Africa, causing complete crop failure in severe cases [5,6,7,8,9]. *Striga hermonthica*, an obligate root parasite of cereals, infests major tropical regions of sub-Saharan Africa, the Middle East, and Asia [10,11,12]. The control of this parasite is very challenging due to high seed proliferation, seed longevity, easy dispersion, a complex life cycle, and wide host adaptability [13,14,15]. As the germination of *Striga* seeds depends on host-released stimulants, mainly strigolactones (SLs), the infestation of this parasite can be lowered by reducing the release of these stimulants in the rhizosphere [16,17,18].

SLs were first identified as germination stimulants for root parasitic weeds, such as *Striga* [19]. However, the most important function of SLs in the rhizosphere is to initiate symbiosis with arbuscular mycorrhizal fungi (AMF) under nutrient deficiency conditions, particularly low phosphate (Pi) availability. Thus, SLs serve as the signals for inducing hypha branching in AM fungi and triggering their metabolism, paving the way for establishing the AM symbiosis that supports the mineral and water supply of the plant and provides the fungus with photosynthetic products [20,21]. Within plants, SL is a plant hormone that regulates several aspects of plant architecture including shoot branching, stem thickness, root development, and leaf senescence, in addition to its involvement in the biotic and abiotic stress response [22,23,24,25].

The evolutionarily conserved SL biosynthetic pathway starts with a reversible isomerization of all-*trans*-β-carotene into 9-*cis*-β-carotene by the isomerase DWARF27 (D27) [23,26] (Figure 1). The resultingly formed 9-*cis*-β-carotene is the substrate of CAROTENOID CLEAVAGE DIOXYGENASE 7 (CCD7) that cleaves it into a 9-*cis*-configured apocarotenal and β-ionone. The former metabolite is the substrate of CCD8 that mediates the formation of the central SL biosynthesis intermediate carlactone (CL), by catalyzing a combination of reactions [27,28,29]. Modifications of CL constitute the second part of SL biosynthesis, which leads to structurally diverse SLs and is catalyzed by cytochrome P450 monooxygenase (CYP) enzymes, particularly from the 711A clade/MORE AXILLARY GROWTH1 (MAX1), and other types of enzymes [23,30,31,32]. In rice, CL is converted by MAX1-900 (Os900) into 4-deoxyorobanchol (4DO), which is hydroxylated into orobanchol (Oro) by MAX1-1400 (Os1400) (Figure 1) [30,33,34]. Both SLs are considered canonical SLs, which are released into the rhizosphere.

Recently, we discovered zaxinone as an apocarotenoid metabolite that promotes plant growth by modulating sugar metabolism and negatively regulates SL biosynthesis in rice [35]. The reduction in SLs upon zaxinone treatment led to low *Striga* infestation under greenhouse conditions [36]. However, the synthesis of zaxinone is rather complex, expensive, and laborious, which is an obstacle for investigating its biology in the laboratory as well as for exploring its potential in agricultural applications. To solve this problem, we developed easy-to-synthesize mimics of zaxinone, namely MiZax3 (MZ3) and MiZax5 (MZ5) (chemical structure shown in Figure 2) as two promising compounds, which showed activities comparable to those of zaxinone in improving growth and reducing SL content in roots and root exudates of rice plants [37,38]. Moreover, these mimics also promoted the growth and yield of horticultural crops under open-field conditions [37,38,39]. Assuming that structural modifications of MiZax3 and MiZax5 might enhance their biological activities, we designed and synthesized MiZax3 (MZ3-1, MZ3-2, MZ3-3, MZ3-4, MZ3-6, MZ3-7, and MZ3-8) and MiZax5 derivatives (MZ5-2 to MZ5-8). In this study, we describe the structure and synthesis of these compounds and the evaluation of their effect on rice growth and *Striga* infestation, in comparison with those of their respective parent compound.

## 2. Materials and Methods

### 2.1. Plants Material and Chemicals

*Striga* seeds were collected from a sorghum field in Sudan (provided by late Prof. Abdel Gabar Babiker, National Center for Research, Khartoum, Sudan). The *Striga*-susceptible rice cultivar IAC-165 used in *Striga* emergence study was obtained from Africa Rice Tanzania (Courtesy of Jonne Roddenburg). Three sets of MiZax were prepared to see the possible effects on growth, SL biosynthesis, and *Striga* infestation in rice. The first and second set included derivatives of MZ3 (MZ3-1 to MZ3-4 and MZ3-6 to MZ3-8) while the third set consisted of derivatives of MZ5 (MZ5-2 to MZ5-8). The parent MZ3 and MZ5 were included as positive controls for comparison. The synthesis scheme of MZ3 and MZ5 derivatives is shown below and physico-chemical properties of all new chemicals are shown in Appendix A.

### 2.2. General Procedure for the Synthesis of MiZax3 Derivatives

The physico-chemical properties of MZ3 derivatives are shown in Appendix A. To prepare MZ3-1, in a round-bottom flask, Sodium borohydride (NaBH_4_) (3.0 mM) was added to the solution of MZ3 [37] (3.0 mM) in an ice-cooled methanol (7.0 mL) and then stirred for 10 min. After stirring at room temperature for 30 min, the reaction was terminated via the addition of saturated Ammonium chloride (NH_4_Cl). Then, ethyl acetate (15 mL) was added to the reaction mixture, which was washed with water and brine. The organic layer was dried over anhydrous sodium sulfate and concentrated under reduced pressure, then purified via column chromatography on silica gel (Wakosil^®^ C-300HG; Fujifilm Wako Pure Chemical Corporation, Osaka, Japan), in which a mixture of Hexane–Ethyl acetate was used as an eluent to give MZ3-1 (Figure 3).

For MZ3-2, in a round-bottom flask, 60% Sodium hydride (NaH) (2.0 mM) was added to the solution of MZ3-1 (1.0 mM) and Iodomethane (2.0 mM) in ice-cooled Tetrahydrofuran (THF) (5.0 mL) and then stirred for 10 min. After stirring at room temperature overnight, the reaction was terminated via the addition of water (15 mL). Then, Ethyl acetate (10 mL) was added to the reaction mixture, which was washed with water and brine. The organic layer was dried over anhydrous Sodium sulfate and concentrated under reduced pressure to give a residue, which was then purified via column chromatography on silica gel (Wakosil^®^ C-300HG), in which a mixture of Hexane–Ethyl acetate was used as an eluent to give MZ3-2 (Figure 3).

For MZ3-3 and MZ3-4, in a round-bottom flask, Methoxylamine hydrochloride (1.2 mM) and Potassium acetate (2.0 mM) were added to the solution of MZ3 (1.0 mM) in ice-cooled Methanol (10 mL) and then stirred for 10 min. After stirring at room temperature overnight, the reaction was terminated via the addition of water (15 mL). Then, Ethyl acetate (10 mL) was added to the reaction mixture, which was washed with water and brine. The organic layer was dried over anhydrous Sodium sulfate and concentrated under reduced pressure to give a residue, which was then purified via column chromatography on silica gel (Wakosil^®^ C-300HG), on which a mixture of Hexane–Ethyl acetate was used as an eluent (Figure 3).

Similarly, for MZ3-6, MZ3-7 and MZ3-8, in a round-bottom flask, the mixture of Dimethyl formamide (2.0 mL), 3-(4-Methoxyphenoxy)benzaldehyde [37] (1.0 mM) and the witting reagent (4.0 mM) was stirred for 2.0 h at 80 °C, and then ethyl acetate (15 mL) was added to the reaction mixture, which was washed with water and brine. The organic layer was dried over anhydrous Sodium sulfate and concentrated under reduced pressure, then purified via column chromatography on silica gel (Wakosil^®^ C-300HG), in which a mixture of hexane–ethyl acetate was used as an eluent. For MZ3-6, 1-(triphenylphosphoranylidene)-2-butanone was used as the Wittig reagent. For MZ3-7, methyl 2-(triphenylphosphoranylidene)acetate was used as the Wittig reagent. For MZ3-8, 3-methyl-1-(triphenylphosphoranylidene)-2-butanone was used as the Wittig reagent (Figure 3).

### 2.3. General Procedure for the Synthesis of MiZax5 Derivatives

The 1st step was as follows: in a round-bottom flask 1,3-dibromobenzene or 3,5-dibromopyridine (2.0 mM), 4-methoxybenzeneboronic acid or 6-methoxy-3-pyridylboronic acid (1.0 mM), 2N Na_2_CO_3_ (4.5 mL) and tetrakis(triphenylphosphine)palladium (0.01 mM) in THF (15.0 mL) were refluxed overnight. After the completion of the reaction, THF was removed via evaporation and the reaction mixture was diluted with Ethyl acetate (EtOAc) and washed with water and brine. The mixture was dried over Sodium sulfate, concentrated, and purified via silica gel column chromatography (Wakosil^®^ C-300HG), eluting it with a hexane–ethyl acetate mixture to give the compound (Figure 4).

The 2nd step was as follows: in a round-bottom flask, each of the three compounds synthesized in the first step, as well as MZ5 (each at 1.0 mM), 3-acetylbenzeneboronic acid or (5-acetylpyridin-3-yl)boronic acid (1.0 mM), 2N Na_2_CO_3_ (4.5 mL), and tetrakis(triphenylphosphine)palladium(0) (0.01 mM) in THF (15.0 mL) were refluxed overnight. After the completion of the reaction, THF was removed and the reaction solution was diluted with EtOAc and washed with water and brine. The product was dried over sodium sulfate, concentrated, and purified via silica gel column chromatography (Wakosil^®^ C-300HG) using Hexane–Ethyl acetate as eluent to give the compounds (Figure 5).

### 2.4. Rice Cultivation Conditions

Firstly, the effect of the new series of MiZax was studied on rice growth and development; then, the effects of these compounds were investigated for SL biosynthesis and *Striga* infestation. For this purpose, rice (Nipponbare) was grown in magenta boxes on agar under controlled conditions. Seeds were surface-sterilized in 50% Sodium hypochlorite solution for 20 min, washed with sterile milliQ water, and germinated in the dark in an incubator at 30 °C in magenta boxes for 48 h. The magenta boxes with pre-germinated seeds were transferred to a growth chamber with standard growth conditions (at 28/25 °C, 12 h day/ 12 h night, and 200 µM photons m^−2^ s^−1^) for one week. For phenotyping, rice seedlings were transferred into 50 mL tubes, and filled with half-strength modified Hoagland nutrient solution (for detail see Appendix A) with an adjusted pH of 5.8. Rice seedlings were treated through root application by transferring them into 50 mL tubes containing the nutrient solution supplemented with MiZax at a 1.0 µM concentration for two weeks. The solution was refreshed in two days intervals.

### 2.5. Striga Germination Bioassays

Assays were performed following the procedure described before [40]. For this purpose, rice plants were grown hydroponically in 50 mL tubes for two weeks under low-phosphate conditions (for detail see Appendix A) and treated with MiZax at a 5.0 μM concentration for 6 h. SLs were collected from exudates using a C_18_ column. Eluates were applied to pre-conditioned *Striga* seeds. For pre-conditioning, *Striga* seeds were surface-sterilized with 50% commercial bleach for 5 min. The bleach was then removed with sterilized milliQ water via six subsequent washings. Seeds were then dried in a laminar flow cabinet, and around 50–100 *Striga* seeds were spread uniformly on 9.0 mm glass fiber filter paper disc. Then, 12 discs with *Striga* seeds were transferred on Whatman filter paper, and moistened with 3.0 mL of sterilized milliQ water in a Petri plate. The Petri plates were sealed with parafilm, wrapped in aluminum foil, and incubated at 30 °C for 10 days. The pre-conditioned *Striga* seeds were treated with each sample (at 55 μL per disc) and incubated again at 30 °C for 24 h. After scanning the discs using a microscope, germinated and total *Striga* seeds were counted using the software SeedQuant V1 [41] to calculate the germination percentage.

### 2.6. Strigolactone Quantification in Root Exudates

For SL analysis, one-week-old rice seedlings were transferred into 50 mL falcon tubes, containing Hoagland nutrient solution with low P (for detail see Appendix A), and allowed to grow in the bio-chamber for 2 weeks. On the day of root exudate collection, the roots of hydroponically grown rice seedlings were treated with compounds at a 5.0 µM concentration for 6.0 h, and then root exudates were collected from each tube for LC-MS/MS analysis. Analysis of SLs in rice root exudates was performed according to the published protocol [42]. Briefly, root exudates spiked with 2.0 ng of GR24 were place on a C_18_-Fast Reversed-Phase SPE column (500 mg/3 mL; GracePure), preconditioned with 3.0 mL of methanol followed by 3.0 mL of water. After washing with 3.0 mL of water, SLs were eluted with 5.0 mL of acetone. Thereafter, the SL-containing fraction was concentrated into a SL aqueous solution (∼500 μL), followed by 1.0 mL of ethyl acetate extraction. Briefly, 750 μL of the SL-enriched fraction was dried under vacuum. The final extract was re-dissolved in 100 μL of acetonitrile:water (25:75, *v*:*v*) and filtered through a 0.22 μm filter for LC-MS/MS analysis. SLs were quantified via LC-MS/MS using UHPLC-Triple-Stage Quadrupole Mass Spectrometer (Thermo Scientific^TM^ Altis^TM^, Waltham, MA, USA) through multiple reaction monitoring (MRM) experiments. The characteristic MRM transitions (precursor ion → product ion) were 331.15 → 216.0, 331.15 → 234.1, 331.15 → 97.02 for 4-deoxyorobanchol; 347.14 → 329.14, 347.14 → 233.12, 347.14 → 205.12, 347.14 → 97.02 for orobanchol; 361.16 → 247.12, 361.16 → 177.05, 361.16 → 208.07, 361.16 → 97.02 for 4-oxo-MeCLA isomer; 333.17 → 219.2, 333.17 → 173.2, 333.17 → 201.2, 333.17 → 97.02 for putative 4-oxo-hydroxyl-CL (CL + 30); 317.17 → 164.08, 317.17 → 97.02 for putative oxo-CL (CL + 14); 299.09 → 185.06, 299.09 → 157.06, 299.09 → 97.02 for GR24.

### 2.7. Striga Emergence in Pots under Greenhouse Conditions

A *Striga* emergence assay in pots under greenhouse conditions was performed as described before [40,43]. About 0.5 L of a mixture of sand and soil (Stender, Basissubstrat) in a 1:3 ratio was added to the bottom of a 3.0 L perforated plastic pot. Then, approximately 8000 *Striga* seeds (about 20 mg) were mixed uniformly in 1.5 L of the soil mixture and added on the top of the pot. The *Striga* seeds in each pot were pre-conditioned for 10 days at 30 °C with light irrigation under greenhouse conditions. Then, three 10-days-old rice seedlings (IAC-165) were planted in the middle of each pot. Each pot was treated with the selected MiZax at a 5.0 μM concentration with 300 mL of a low-P nutrient solution twice a week, for up to 8 weeks. Rice plants were allowed to grow under normal growth conditions (30 °C, 65% RH). After 70 days of sowing, *Striga* emergence was determined in each pot and compared with that of the blank treatment.

### 2.8. Gene Expression Analysis

Roots of rice seedling were ground and homogenized in liquid nitrogen. Total RNA was isolated using a Direct-zol RNA Miniprep Plus Kits (ZYMO Research, Irvine, CA, USA), and cDNA was synthesized from 1.0 µg of total RNA using iScript cDNA Synthesis Kit (BIO-RAD Laboratories, Hercules, CA, USA) according to the manufacturer’s instructions. Gene expression levels were detected via real-time quantitative RT-PCR (qRT-PCR) which was performed with SsoAdvanced™ SYBR^®^ Green Supermix (BIO-RAD Laboratories, CA, USA) in a CFX384 Touch™ Real-Time PCR Detection System (BIO-RAD Laboratories, CA, USA). The primers used for qRT-PCR analysis are listed in Appendix A. Gene expression levels were calculated via the normalization of a housekeeping gene in rice, ubiquitin (OsUBQ) (Appendix A). The relative gene expression level was calculated in accordance with the 2^−ΔΔCT^ method.

### 2.9. Statistical Analysis

Collected data were analyzed statistically using statistical software package R (version 3.2.2) and Graphpad prism (version 9.1.1). One-way analysis of variance (ANOVA) was used for analyzing the data.

## 3. Results

### 3.1. Effect of MZ3 and MZ5 Derivatives on Rice Growth and Development

To assess the effect of modifying the ketone group on biological activity, we designed and synthesized various MZ3 derivatives, in which we replaced the ketone group with a hydroxyl, methoxy-, or imine group (for structures, see Figure 2), and evaluated their growth effects at a 1.0 µM concentration on hydroponically grown rice seedlings. The first set of MZ3 derivatives (MZ3-1 to MZ3-4) showed significant growth-promoting effects (Figure 6B–D and Figure 7A). We observed an increase in root length from 22% to 47%, in crown root number from 17 to 53%, and in total dry biomass of around 19–45%. Although we also observed a positive effect on shoot length (Appendix A); the impact of MZ3-1 to MZ3-4 on roots was comparably more pronounced. Overall MZ3-1 outperformed concerning root length (47%), while MZ3-2 showed a higher increase in crown root number (53%) and total dry biomass (54%). The parent compound MZ3 showed 37%, 25%, and 28% increases in root length, crown root number, and total dry biomass, respectively.

Similarly, we evaluated the activity of MZ3-6, MZ3-7, and MZ3-8, in which we extended the chain after the carbonyl group in MZ3 or replaced it with a methyl carboxy group. We observed a 25% to 38% increase in root length, an increase of 9–14% in crown root number, and an increase of 2–16% in total dry biomass (Figure 6E–G and Figure 7). MZ3-7 showed the best performance with respect to the increase in crown root number (14%) and biomass (15%), while MZ3-6 caused a 8% reduction in root biomass. Similarly, to that in MZ3-1 to -4, the impact of these compounds was more pronounced on roots than shoots. The parent compound MZ3 showed better activity on root length, compared to MZ3-6 to -8, but showed weaker activity for crown root number and root biomass.

Moreover, to evaluate the bioactivities of (a) N-containing heterocycle(s), we synthesized a series of MZ5 derivatives (MZ5-2 to MZ5-8), in which we replaced phenyl groups with (a) N-containing heterocycle(s). The biological activity of these new MiZax was evaluated, following the workflow used for the other compounds. Most of the MZ5 derivatives showed significant effects on rice root growth and development (Figure 6H–J and Figure 7C). We observed an increase in root length, in a range from 9% to 18%, and in root dry biomass of around 6–31%; however, we also observed some negative effects on crown root number of some of the MZ5 derivatives, especially MZ5-7 and MZ5-8, which caused around a 10% decrease compared to the blank treatment. The developed MZ5 derivatives did not show any significant effect on shoots (Appendix A).

### 3.2. Effect of MZ3 and MZ5 Derivatives on SL Biosynthesis

Next, we determined the effect of MZ3 and MZ5 derivatives on SL release. Treatment with these compounds caused a significant reduction in SL content, as confirmed via LC-MS/MS analysis (Figure 8). Upon the application of MZ3-1 to -4, we observed 28–64% reductions in orobanchol (Oro), 31–85% reductions in 4-Deoxyorobanchol (4DO), 26–52% reductions in putative oxo-carlactone (oxo-CL) [44], and 75–84% reductions in methyl 4-oxo-carlactonoate (4-oxo-MeCLA, a putative non-canonical SL [30]) (Figure 8B–E). Similarly, we observed 41–70% decreases in Oro content upon the application of MZ3-6, -7, and -8, while 4DO, the second canonical SL in rice, was suppressed only by MZ3-6. MZ3-6 and MZ3-7 also caused 55–59% reductions in the level of oxo-CL. Interestingly, the application of MZ3-8 increased the content of oxo-CL by more than 100% (Figure 8H). The positive control, the parent compound MZ3, showed a 52% reduction in Oro, a 40% reduction in 4DO, a 45% reduction in oxo-CL, and a 85% reduction in 4-oxo-MeCLA (Figure 8F–I). Furthermore, we observed 34–78% decreases in the content of 4-oxo-MeCLA. In another study, we examined the effect of MZ5-2 to MZ5-8 on SL release. Most of the MZ5 derivatives showed significant reductions in the content of different SLs (Figure 8J–M). We observed a 31–79% reduction in Oro, except for the case of MZ5-6. We also observed a 2–38% decrease in 4DO, a 30–77% reduction in oxo-CL, and a 32–49% decrease in 4-oxo-MeCLA caused by the MZ5 derivatives.

Explaining the reduction in SL content, the application of MZ3-6 to MZ3-8 caused a decrease in the transcript level of genes mediating SL biosynthesis, i.e., *D27*, *CCD7*, *CCD8*, and *MAX1-900*, compared to that with the blank treatment (Figure 9).

### 3.3. Effect of MZ3 and MZ5 Derivatives on Striga Germination

The suppression of SL biosynthesis by MiZax was expected to ultimately result in the reduction of germination efficiency of *Striga* seeds, due to the altered composition of rice root exudates. Indeed, the effect of the treatments with MZ3-1 to -4 on *Striga* seed germinating efficiency was less clear and differed depending on the compound (Figure 10B). While we observed 10 and 14% reductions with MZ3-1 and 3-4, respectively, and the application of MZ3-2 caused a negligible reduction, MZ3-3 caused an increase in *Striga* seed germinating efficiency (Figure 10B). Treatment with the positive control MZ3 led to a 28% reduction in *Striga* germination. The application of MZ3-6 and MZ3-8 caused a 16–25% reduction in *Striga* seed germination, compared to that with the blank treatment (Figure 10C). The parent compound MZ3 showed a 26% reduction in *Striga* seed germinating efficiency. Similarly, we observed a considerable reduction in *Striga* germination with some of the MZ5 derivatives. One of the derivatives, MZ5-3, showed a 34% reduction in *Striga* germination, exhibiting similar bioactivity to that of its parent, MZ5, with 36% reduction in *Striga* germination, whereas MZ5-6 showed the smallest reduction in *Striga* germination efficiency (Figure 10D).

### 3.4. Effect of Selected MiZax on Striga Emergence and Host Growth in Pots under Greenhouse Conditions

To determine their effect on *Striga* emergence under greenhouse conditions, we selected MZ3-1, MZ3-6, and MZ5-2 as a candidate from each of the three sets of compounds (Figure 11). We observed a clear reduction in *Striga* emergence, ranging from 21 to 32%, upon the application of the selected new MiZax derivative. The application of MZ5-2 led to a 32% reduction, followed by MZ3-1 with a 22% and MZ3-6 with a 21% reduction. With about a 40% reduction, the parent compound MZ3 showed significantly higher activity than the new compounds did, while, surprisingly, that of MZ5 was weaker than that of others (a 19% reduction). The reduction in *Striga* infestation also led to the better growth of rice plants that showed 52% increase in plant height upon MZ3 treatment and 18% increase upon MZ3-1 treatment. However, the host plants in MZ5-2- and MZ3-6-treated pots did not show considerable improvements in plant height.

## 4. Discussion

### 4.1. Rice Growth Improves with the Application of the New MiZax

Recently, we reported on the growth-promoting effect, particularly on root growth and development, of zaxinone and its mimics MiZax3 and 5 [35,37,38,39]. The encouraging results obtained via the corresponding studies indicated the utility of these bio-stimulants for field application to enhance crop productivity [37,38,39]. In the present study, we developed a new series of zaxinone mimics via the structural modification of MiZax3 and MiZax5 to possibly enhance their bioactivity. In the first series of MZ3 derivatives, MZ3-2 outperformed all other MiZax with respect to the effect on crown root number and root biomass. In the second set of the MZ3 derivatives (MZ3-6 to MZ3-8), MZ3-7 showed higher activity in enhancing the number of crown roots and increasing total root biomass in rice seedlings, compared to that with the blank treatment and MZ3. In the third series, the MZ5 derivative MZ5-6 showed a very promising effect on root length and root biomass. The structural modifications of MiZax demonstrated an application possibility of these modified molecules for particular modulations in root growth, which might have a general effect on plant growth. These data indicate that extending the chain after the ketone group in MZ3 or replacing the ketone with a methoxy group enhanced root growth-promoting activity. In addition, the higher activity of MZ5-6 with respect to increased root length and biomass revealed a specific effect of replacing the two phenyl groups adjacent to the ketone group in MZ5 with N-containing heterocycles.

### 4.2. Strigolactone Biosynthesis Is Negatively Regulated by the New Series of MiZax

Carotenoid-derived plant hormone SLs are released by plant roots into the soil [23,45]. These small molecules not only regulate developmental processes to fine-tune shoot and root architecture to overcome environmental changes [46], but also enable symbiotic fungi [20] to detect their host plants [19,47]. However, the amount and pattern of SLs released by the host roots in the rhizosphere directly correlate with the infestation rate by root parasitic weeds [48,49]. Therefore, by reducing plant-secreted SLs in the rhizosphere, we can possibly lower infestation and attack by these weeds [48].

Among the here-synthesized MiZax, MZ3-2, MZ3-6, and MZ5-3 caused maximum reductions in the level of various SLs, particularly Oro and oxo-CL. In fact, MZ3-6 was at least comparable with the parent compound MZ3 in down-regulating the expression of SL biosynthetic genes (*D27*, *CCD7*, *CCD8*, and *MAX1-900*) in rice seedlings. It is worthy of note that the effect of the MiZax derivatives differed depending on the type of SLs, indicating their importance, particularly for lowering *Striga* infestation.

### 4.3. Reduction of Striga Infestation by the Application of the New Series of MiZax

Witchweeds (*Striga* spp.) and broomrapes (*Orobanche* and *Phelipanche* spp.) are the two most destructive groups of parasitic weeds, causing huge losses in agriculture [4,17]. Since they have a strong dependency on host-released SLs for seed germination [45,47], their infestation can be lowered by reducing the levels of released SLs [48,50,51]. We have already shown that zaxinone and MiZax reduced *Striga* infestation through the negative regulation of SL biosynthesis in rice as a host plant [35,37]. Herein, we tested the *Striga* seed germinating efficiency of root exudates from hosts treated with new MiZax. In the first set, i.e., MZ3-1 to MZ3-4, we only observed reductions in *Striga* germination with MZ3-1 and MZ3-4, which were also weaker than that of MZ3. It is important to note that MZ3-3 even showed a positive effect on *Striga* seed germination, indicating that it might induce or support the germination process in *Striga* seeds and ultimately might enhance the *Striga* infection rate in the host. MZ3-1, MZ3-2, and MZ3-4 were not efficient in reducing the stimulation of *Striga* germination by rice root exudates although resulted in a significant reduction in SL content. In contrast, the structural modifications of MZ3, which led to MZ3-6 and MZ3-7, did not significantly affect the ability of this compound to reduce the seed-germinating activity of root exudates, while that of MZ3-8 was significantly weaker than that of MZ3, which is in line with the positive effect of MZ3-8 on the content of some SLs. Among the MZ5 derivatives, MZ5-3 showed a maximum reduction in *Striga* seed germination, comparable to that with MZ5 itself. The other MZ5 derivatives were generally weak with respect to this activity. The results obtained in the greenhouse experiment indicate that MZ3 is still the most useful zaxinone mimic in alleviating *Striga* emergence.

### 4.4. Possible Reasons for Differences Observed in the Bioactivity of the New MiZax

Previously, we showed that MiZax1, containing a shorter carbon chain, failed to lower SL release, indicating the importance of chain length between the ketone group and phenyl ring [37]. In this study, we assessed the effect of extending the chain after the ketone group, substituting the ketone with other functional groups (imine, alcohol, or ether), or replacing the phenyl group(s) with (a) nitrogen-containing heterocycle(s). Herein, substituting the ketone with an imine, alcohol, and ether group (MZ3-1 to MZ3-4) led as in the case of MZ3-2 to improved growth-promoting activity. Notably, the structural modifications in some of the MZ3 derivatives caused a reduction in *Striga* seed germinating efficiency. It might be speculated that some of the MZ3 derivatives could interfere with the germination process in *Striga*. Similarly, when extending the chain (MZ3-6, MZ3-7, and MZ3-8), one of these compounds showed better activity for root growth and reduction in SL biosynthesis. In the case of MZ5 derivatives, replacing the phenyl group(s) with (a) nitrogen-containing heterocycle(s) improved the activity in some of the MZ5 derivatives such as MiZax5-3 particularly in lowering *Striga* infestation.

## 5. Perspectives, Future Outlook and Conclusions

The employment of bio-stimulants, supplements, and organic manuring is always recommended to improve plant growth and soil fertility [52,53,54]. In this work, we designed and developed 14 new biostimulants via the structural modification of the two highly efficient mimics MiZax3 or MiZax5 for possible improvements in their bioactivity, but these new compounds did not outperform their parent compounds, particularly MZ3, which was highly efficient in promoting growth, regulating SL biosynthesis and alleviating *Striga* seed germination. However, the new set of MiZax contained compounds that showed better activity in certain aspects, such as an increase the number of crown roots. Moreover, we observed that some of the newly developed compounds showed contradictory effects on the content of particular SLs. For instance, MZ3-8 application led to a reduction in orobanchol and an increase in 4-deoxyorobanchol content but interestingly, its application increased the content of oxo-CL by more than 100%. Understanding the molecular background of the observed differences requires further studies at the molecular and cellular level, which is currently hampered by the lack of knowledge about zaxinone/MiZax receptor/binding protein(s). Indeed, the complex mechanisms underlying the positive regulatory effects of MiZax on plant growth and development as well as on the negative regulation of SL biosynthesis remain elusive. At this stage, we further suggest using the parent MiZax, especially MiZax3, for improving plant growth and controlling *Striga* infestation. We are currently optimizing the application protocol by defining the optimal concentration/dose of MiZax, and the method of application (soil vs. foliar). Further studies are also needed to determine the fate of MiZax after application, its persistency, and its impact on soil microbes.

Taken together, our results demonstrate that structural modification of MiZax3 and MiZax5 can change bioactivity. Although the newly developed compounds did not outperform their parental compounds, our data pave the way for their further modification for possible improvement in MiZax activity.

## Figures and Tables

**Figure 1 biomolecules-13-01206-f001:**
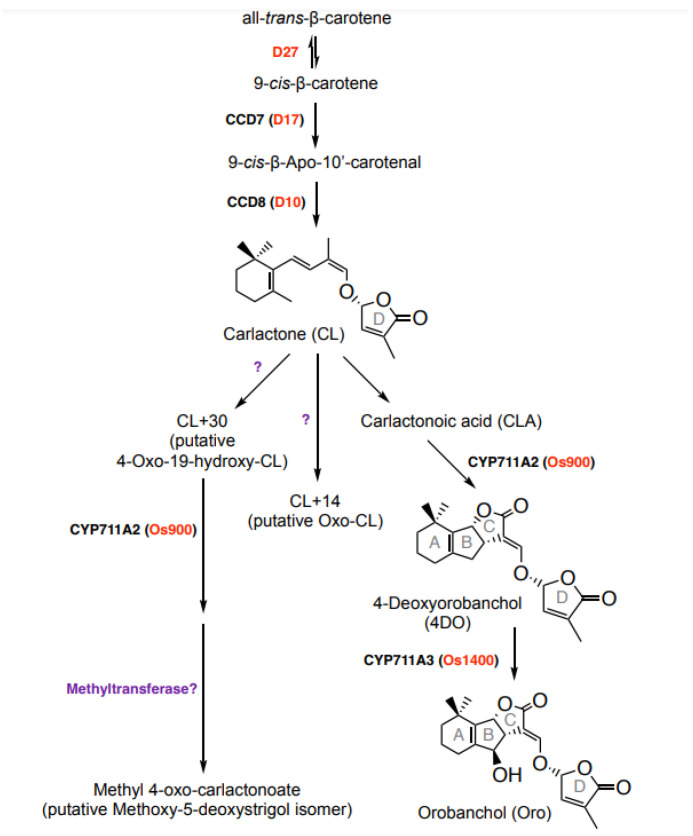
Strigolactone biosynthetic pathway in rice. SL biosynthesis involves sequential actions of an isomerase, DWARF27 (D27); two CAROTENOID CLEAVAGE DIOXYGENASES 7 (CCD7) and CCD8; cytochrome P450 (CYPs) from the 711 clade and other enzymes. D27 catalyzes the reversible isomerization of all-*trans*- into 9-*cis*-β-carotene. The latter one is further converted by CCD7 and CCD8 into carlactone (CL), a key intermediate in SL biosynthesis. Canonical and non-canonical SLs arise through the conversion of CL by different CYPs and further modifications by other enzymes, such as a methyltransferase.

**Figure 2 biomolecules-13-01206-f002:**
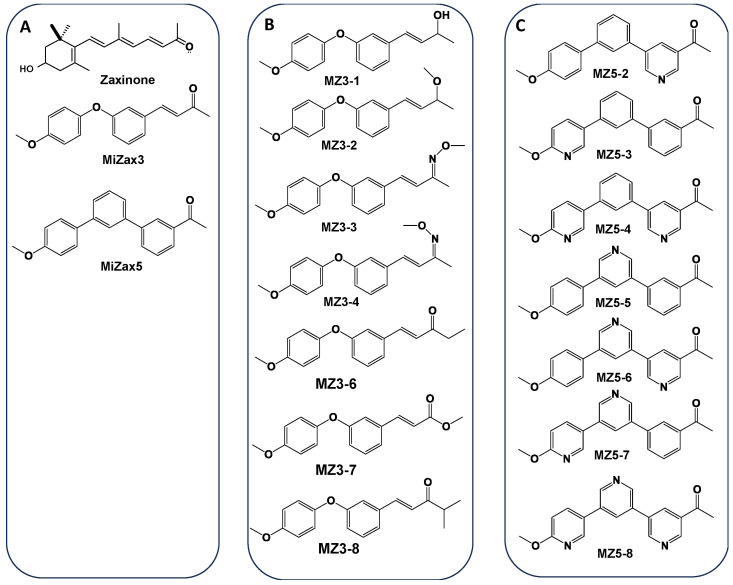
Structure of the mimics of zaxinone (MiZax). (**A**) Structure of zaxinone, MiZax3 and MiZax5. (**B**) Structure of the MiZax3 derivatives MZ3-1, MZ3-2, MZ3-3, MZ3-4, MZ3-6, MZ3-7 and MZ3-8. (**C**) Structure of the MiZax5 derivatives MZ5-2, MZ5-3, MZ5-4, MZ5-5, MZ5-6, MZ5-7, and MZ5-8.

**Figure 3 biomolecules-13-01206-f003:**
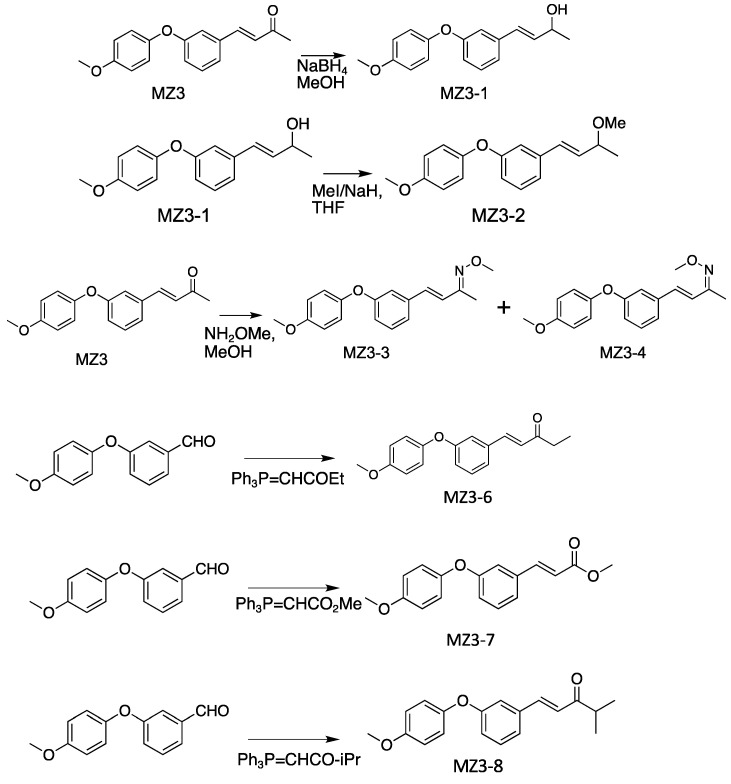
Synthesis of MiZax3 derivatives.

**Figure 4 biomolecules-13-01206-f004:**
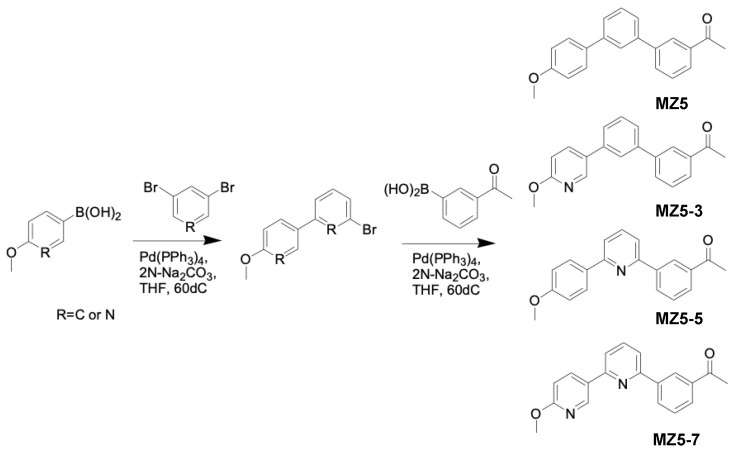
Synthesis of MiZax5 derivatives (MZ5-3, MZ5-5 and MZ5-7).

**Figure 5 biomolecules-13-01206-f005:**
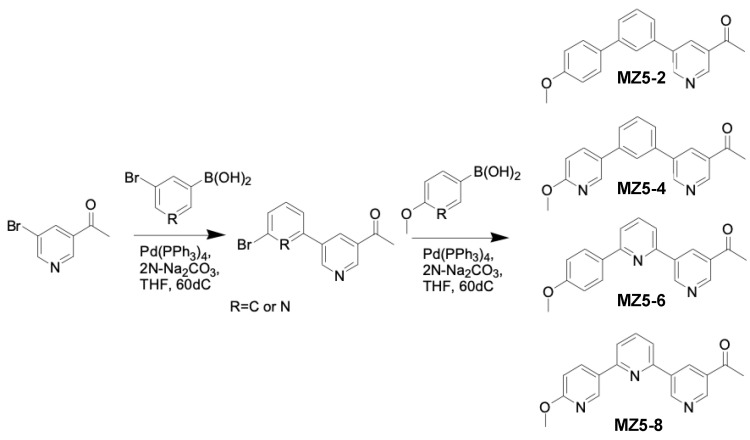
Synthesis of MiZax5 derivatives (MZ5-2, MZ5-4, MZ5-6 and MZ5-8).

**Figure 6 biomolecules-13-01206-f006:**
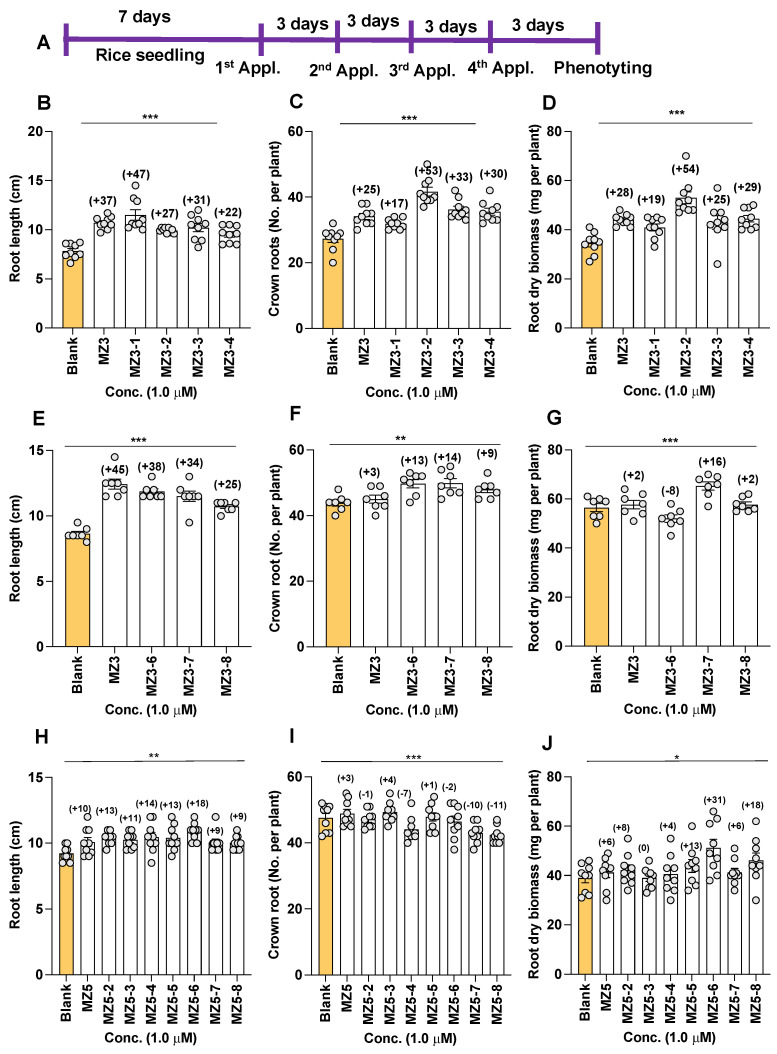
Effect of MZ3 and MZ5 derivatives on root length, crown root number, and root dry biomass in rice. (**A**) Experimental scheme. The compounds were applied at 1.0 µM to hydroponically grown rice seedlings for two weeks. (**B**–**D**) Effect of MZ3-1 to MZ3-4. (**E**–**G**) Effect of MZ3-6 to MZ3-8. (**H**–**J**) Effect of MZ5-2 to MZ5-8. Data are means ± SE (*n* = 7). Values on the top of each bar show a percent increase (+) or decrease (−) over the control treatment. Asterisks denote significance (one-way ANOVA; * *p* < 0.05, ** *p* < 0.005, *** *p* < 0.0005).

**Figure 7 biomolecules-13-01206-f007:**
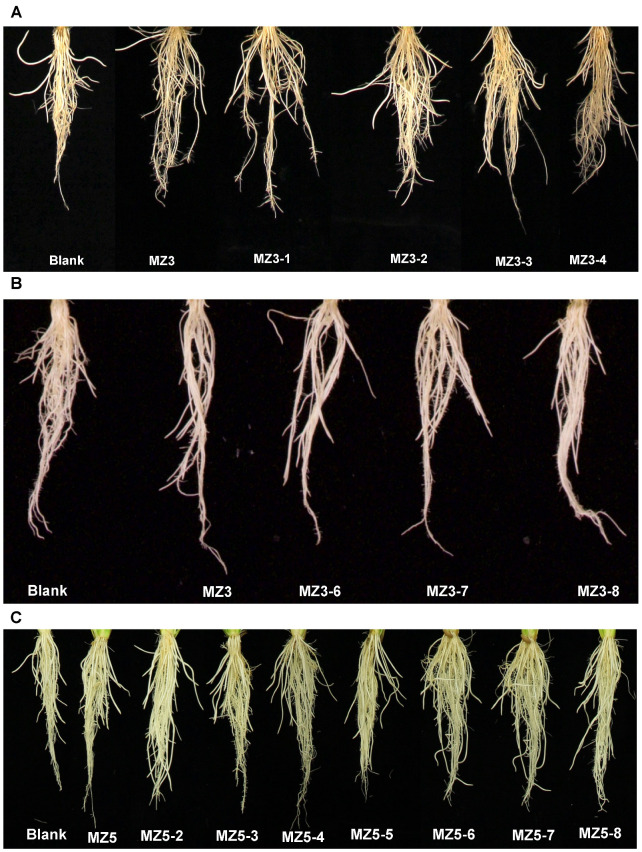
Effect of newly modified MZ3 and MZ5 derivatives on rice root growth. (**A**) The phenotype of roots of rice seedlings in response to MZ3-1 to MZ3-4 application. (**B**) The phenotype of roots of rice seedlings upon application of MZ3-6 to MZ3-8. (**C**) The phenotype of roots of rice seedlings in response to application of MZ5 MZ5-2 to MZ5-8. The newly modified compounds were applied at 1.0 µM to hydroponically grown rice seedlings for two weeks.

**Figure 8 biomolecules-13-01206-f008:**
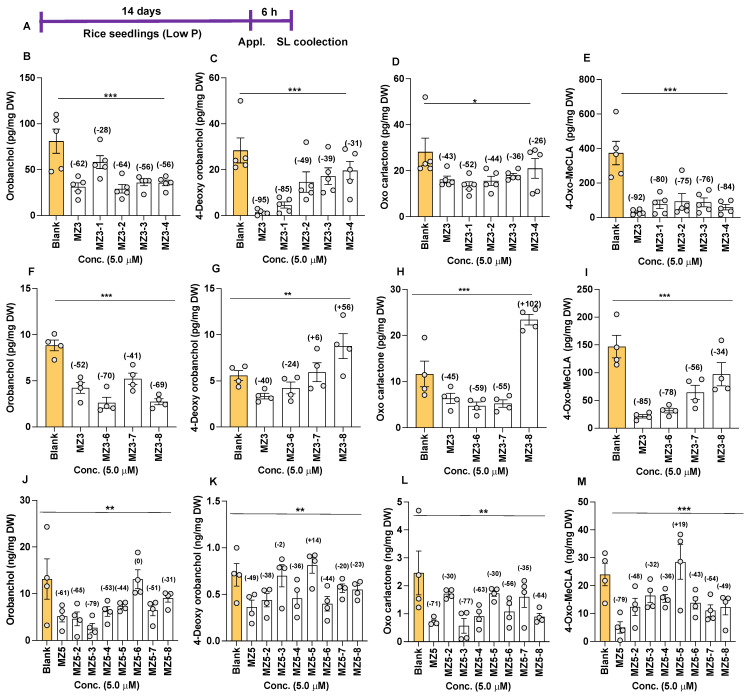
Effect of MZ3 and MZ5 derivatives on SL content of root exudates. (**A**) Scheme of the experiment. The compounds were applied at 5.0 µM concentration for 6.0 h to 2-week-old rice seedlings, grown hydroponically under low-phosphate conditions. The SLs orobanchol, 4-Deoxyorbanchol, oxo-carlactone, and methyl 4-oxo-carlactonoate (4-oxo-MeCLA) were quantified via LC-MS/MS. (**B**–**E**) Effect of MZ3-1 to MZ3-4. (**F**–**I**) Effect of MZ3-6 to MZ3-8. (**J**–**M**) Effect of MZ5-2 to MZ5-8. Data are means ± SE (*n* = 4). Values on the top of each bar show a percent decrease (−) or increase (+) over the blank treatment. Asterisks denote significance (one-way ANOVA; * *p* < 0.05, ** *p* < 0.005, *** *p* < 0.0005).

**Figure 9 biomolecules-13-01206-f009:**
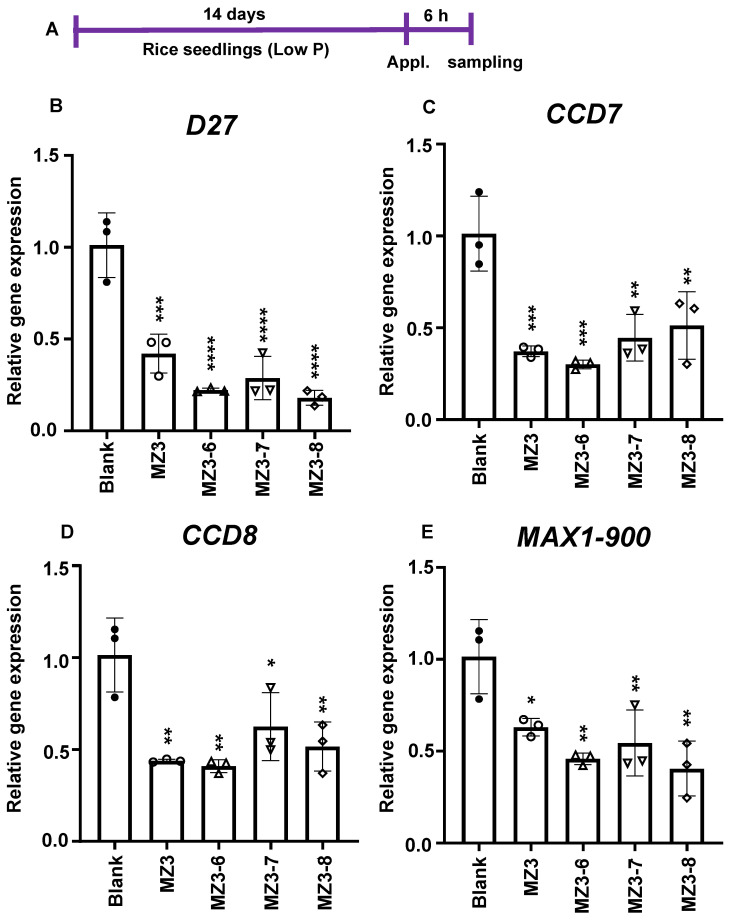
Effect of MZ3 derivatives on gene expression associated with strigolactone biosynthesis. (**A**) Scheme of experiment. (**B**–**E**) Gene expression of *D27*, *CCD7*, *CCD8*, and *MAX1-900* upon the application of MZ3-6 to MZ3-8. The newly designed MiZax compounds were applied at 5.0 µM for 6.0 h to 2-week-old rice seedlings grown hydroponically under low-P conditions. Data are means ± SE (*n* = 3). Asterisks denote significance (one-way ANOVA; * *p* < 0.05, ** *p* < 0.005, *** *p* < 0.0005, and **** *p* < 0.00005).

**Figure 10 biomolecules-13-01206-f010:**
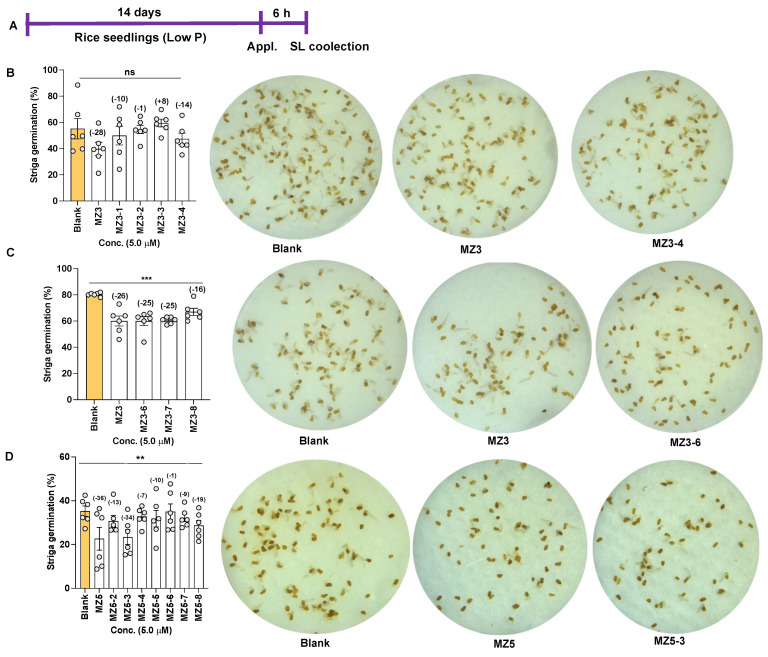
Effect of MZ3 and MZ5 derivatives on *Striga* seed germination. (**A**) Scheme of the experiment. The compounds were applied at 5.0 µM for 6.0 h to 2-week-old rice seedlings grown hydroponically under low-phosphate conditions. Exudates were then collected and applied to *Striga* seeds. (**B**) Germination ratios after the application of MZ3-1 to MZ3-4, (**C**) MZ3-6 to MZ3-8, and (**D**) MZ5-2 to MZ5-8. Data are means ± SE (*n* = 4). Values on the top of each bar show a percent decrease (−) or increase (+) over the blank treatment. Asterisks denote significance (one-way ANOVA; ** *p* < 0.005, *** *p* < 0.0005, ns: non-significance).

**Figure 11 biomolecules-13-01206-f011:**
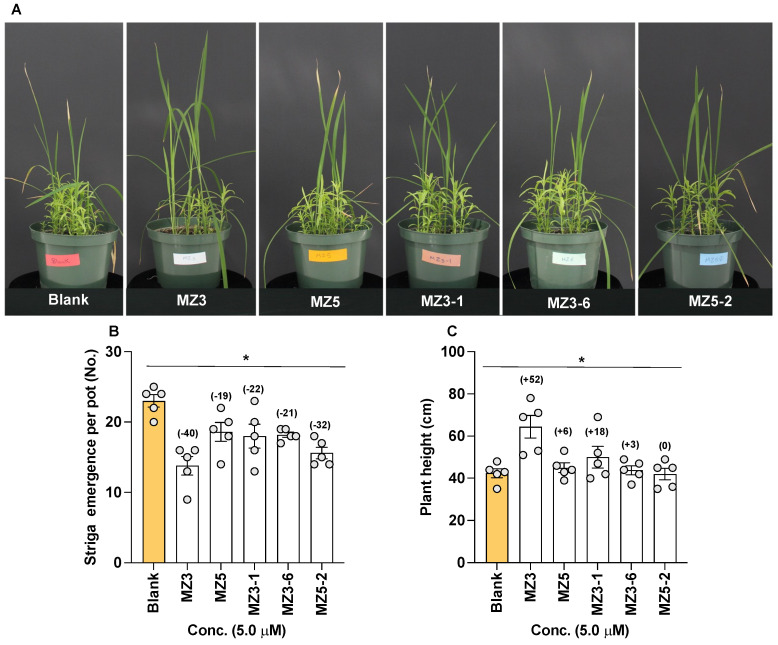
Effect of MiZax on *Striga* emergence in rice. (**A**) Pictures of *Striga* emergence in treated pots. (**B**) Number of emerging *Striga* plants in response to MiZax applications. (**C**) Height of plants in response to MiZax application. Data are means ± SE (*n* = 5). Values on the top of each bar represent a percent decrease (−) or increase (+) over the blank treatment. Asterisks denote significance (one-way ANOVA; * *p* < 0.05).

## Data Availability

The data presented in this study are available on request from the corresponding author.

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
