# Peer review of "New Series of Zaxinone Mimics (MiZax) for Fundamental and Applied Research"

_biomolecules, 2023, doi:10.3390/biom13081206_

Round 1

Reviewer 1 Report

Dear Authors,

I have reviewed your manuscript "New series of zaxinone mimics (MiZax) for fundamental and applied research", submitted for publication in Biomolecules.

In your paper you presented a comprehensive research on the effects of novel MiZax compounds on rice plant growth, the inhibition of strigolactone biosynthesis and the infestation of rice plants with Striga hermonthica. Unfortunately, although some of the 14 newly synthesized compounds displayed a moderately stronger effect on individual parameters of rice plant growth or Striga infestation, none of them seems to overall outperform the effects of the highly efficient MiZax3 and MiZax5. In other words, if I was a rice farmer and had to pick a MiZax compound to treat my plants with, based on your results I would still pick MiZax3 or MiZax5 over any of the newly synthesized compounds you presented in your new paper.

The fact that you did not prove a consistently superior performance of any of the 14 newly synthesized compounds compared to MiZax3 and MiZax5, does not discredit your research in my opinion - your work proves that subtle structural modifications to the MiZax compounds do not importantly improve their agronomical performance, and might suggest, that expectations from novel compounds obtained through further structural modifications of MiZax should be lowered. However, although this conclusion is obvious to the reader after seeing the results of your current research, you did not point it out. Although you compared the effects of the novel MiZax with the previously available MiZax3 and MiZax5 throughout the Results section, your Discussion and Conclusions are focused on praising the novel MiZax compounds and their biological effects, comparing them mostly to untreated (control) plants, as if the (mostly better-performing) MiZax3 and MiZax5 were not already available. The Discussion section mostly relies on vague statements about the effects of structural modifications of MiZax on enhancing their performance, but the particular effects of structural modifications are never discussed - and neither are the novel compounds ever compared to MiZax3 and MiZax5 in terms of overall performance.

My opinion on your paper is that you should completely refocus the tone of your discussion, and offer a more critical comparison between the novel MiZax compounds and their "parent compounds" MiZax3 and MiZax5. I would urge you, when revising your paper, please do not keep the deleted text in tracked changes, but instead only highlight the new portions of text which have been added or revised. For more detailed remarks on your paper, please follow my comments as given below:

·         General language concerns:

o    please put Striga in italic letters throughout the manuscript text

o    please use the word "infestation" instead of "infection" throughout the manuscript text. "Infection" is used for microorganisms, whereas "infestation" is more appropriate for macroscopic organisms such as parasitic pests or weeds.

·         Introduction:

o    line 32: The term "food insecurity" already encompasses the risks of human starvation and malnutrition. Please revise.

o    line 35: sub-Saharan (a letter A is missing)

o    line 55: "The evolutionarily-conserved SL biosynthetic pathway..." - please start a new paragraph with this sentence. The description of the biosynthetic pathway is very complex and should have a whole paragraph dedicated to it to help the reader maintain focus.

o    line 58-59 and elsewhere: Please check the text for a possible presence of double-spaces.

o    line 62: "...and is catalyzed by..." - please replace "is" with "are"

o    line 69: please delete "Scheme of" from the Figure caption

o    Figure 2 and line 76: Please also provide the structure of zaxinone in Figure 2

o    line 78: the original work in which zaxinone was discovered (Wang et al. 2019 published in Nature Communications: https://doi.org/10.1038/s41467-019-08461-1), should be provided here, instead of ref.35.

o    lines 86-94: an explanation regarding the nomenclature of MiZax6, 7, and 8 vs. MiZax3-1, 3-2, 3-3, and 3-4 should be provided here. All these seven compounds have been derived from MiZax3, but only four of them (MiZax3-1 through 3-4) have been named after MiZax3, whereas MiZax6, 7, and 8 have received "names of their own". In which aspect do these two groups of MiZax3 derivatives differ from each other, and why were different nomenclature rules applied to them? A brief explanation should be provided within the Introduction section. Alternatively, if there is no good explanation but these names have been given ad hoc, I would suggest to rename MiZax6, 7, and 8 to MiZax3-6, 3-7, and 3-8, respectively, and to revise, accordingly, the entire manuscript.

o    also lines 86-94: a clear statement of your research goal is missing from the end of Introduction. Why did you synthesize novel MiZax compounds and test their performance in terms of rice plant growth promotion, strigolactone synthesis, and infestation with Striga? Highly efficient MiZax3 and MiZax5 have already been synthesized and tested for these effects, with very satisfying results. Why would you need novel compounds and test them? My guess is that you wanted to investigate whether, through a diverse set of structural modifications of the highly efficient compounds MiZax3 and MiZax5, you could obtain novel compounds with even greater biological activity in terms of the above-mentioned desired effects. This research goal should be more clearly stated at the end of your Introduction, and the Discussion and Conclusions to your work should be formulated so as to position your obtained results in relation to whether or not this research goal has been met, to what extent, and how; or why not.

·         Materials & Methods:

o    line 103: Please provide the affiliation of the Late Prof. Babiker.

o    line 106: includeD

o    line 110: The synthesis scheme and the physico-chemical properties of the 14 newly synthesized MiZax compounds are not provided in Figure 2. Figure 2 only shows their structural formulae. By contrast, the synthesis scheme and the physico-chemical properties of the MiZax compounds are presented in the continuation of the M&M section, which is very useful, detailed and informative in terms of content, but on the other hand it is very poorly structurally defined, in terms of what subsection it belongs to, and whether it consists of text, of figures, or of combination of text and figures. This part of the manuscript (pages 4-9 and start of page 10) calls for clearer organizations into sections, subsections, sub-subsections and/or Figures - or, it can be entirely transferred to the Supplementary material. In any case, it cannot stay the way it is organized right now, because its relation to the manuscript structure is completely unclear.

o    line 112: What aldehyde?

o    line 201 and 209: "2N-Na2CO3" - what does the dash ("-") between "2N" and "Na2CO3" stand for? Did you mean "2N Na2CO3"? Please double-check, and revise if necessary. Same comment applies to the chemical reaction schemes in line 214.

o    page 7, other comments: please fully spell out THF and AcOEt at first mention, as well as all other abbreviations for compounds where standard IUPAC names are not used.

o    line 279: please rename the title of section 2.3 to "Rice Cultivation Conditions". "Details of Experiments" really does not mean anything.

o    line 289-290: Seedlings were treated with MiZax how? Transferred to fresh nutrient media containing MiZax, or did they have the MiZax solution poured on top of them? Please clearly describe the details of the experimental procedure within the M&M text.

o    line 293-294: please define the "low phosphate conditions"

o    line 294-295: If I understand correctly, the SLs were not "collected from root exudates", but instead, the root exudates were collected using a C18 column, and the SLs were isolated from the root exudates, as described in Section 2.5. Please be precise and clear in your statements.

o    line 295: please add "as described further" to the end of the sentence (after "Striga seeds"), to prepare the reader that the procedure is actually about to be explained in the immediate follow-up.

o    line 307: please add the word "rice" (RICE seedlings)

o    line 308: What is the bio-chamber now? I do not recall you introducing a bio-chamber to the reader.

o    Section 2.6: you used capital L for "liter" in this section, whereas you used lowercase "l" everywhere else in the manuscript. Please stick uniformly to either capital L or lowercase "l" (I must say that I personally prefer the capital L) throughout the entire manuscript.

o    please add an additional subsection to Materials & Methods, to describe the software used for statistical analysis.

·         Results:

o    General remark about the Results section: It is very tiring for the reader to read three sets of very similar results one after another. The Authors should consider reorganizing the entire Results section to pull together all the results for all the 14 novel MiZax compounds, either in single graphs, or at least in separate graphs but presented immediately one after another, so that the reader would not jump back and forth between plant growth parameters, SL composition and the Striga germination parameters. Thus, the Figures 3, 6, and 8 could be combined into a single Figure (or they could remain as they currently are, but grouped together one after another). For Figures 4, 7, and 9, I suggest that the results referring to the SL composition in root exudates should be separated from the results referring to Striga germination (they should be placed in separate Figures, and also, all the results referring to the SL composition should be presented before all the results for Striga germination, for all the MiZax compounds).

o    Also, in their narration of the Results, the Authors should stick strictly to the order in which the results are presented in the Figures. Currently, in the paragraph 360-374, the Authors are jumping back and forth between narrating the results presented in the Figures 4 and 5.

o    The subfigure "A" in Figures 3 and 4 should be made a lot larger (at least as large as it is in the case of Figure 5) and more clearly spatially separated from the subfigures B, C, and D.

o    Figure captions: the letters p (for confidence intervals) and n (for sample size) should be written in italic letters in all the figure captions throughout the manuscript.

o    Figures 3-9 (except for Figure 5): in the Figure captions, you state that the "asterisks denote significance determined through one-way ANOVA". However, each asterisk is applied to the entire group of results (for instance, in Figure 3B, the three asterisks mean that the differences within the group are statistically significant at the p < 0.005 level). I find that the Fisher's LSD (least significant differences) test for the determination of the statistical significance of the differences between each treatment and the blank control would be much more informative. Alternatively, instead of comparing each of the MiZax compounds with the blank control, you might want to compare it to the effect of the "parental" MiZax compound (for instance, MiZax3-1 with MiZax3), because this is what is really of greatest interest within your current research.

o    Speaking of control, you have used terms "blank", "mock" and "control" even within the same Figure (for instance, in Figures 4, 7, and 9). Please pick just one of these synonyms ("blank", "mock", or "control") and use it consistently throughout your manuscript, including within all the Figures and Figure captions.

o    line 365: You should turn the reader's attention to the quite remarkable >100% increase in oxo-carlactone in response to MiZax8 (Figure 4D).

o    line 394: MZ3-4, not "MZ-4"

o    in the Results section (for instance, lines 408, 411-412) you use the term "germinating activity" for "germination efficiency". Please revise, and thoroughly check throughout the entire manuscript text.

o    line 457: the maximum, 36% reduction in Striga germination was obtained for the original compound MZ5, from which the other MiZax5s were derived. Thus, the MZ5 derivatives showed a reduction in Striga germination of up to 34% (for MZ5-3).

o    line 459: not "the same bioactivity", but "similar bioactivity".

o    Section 3.4 and Figure 10: The results in Figure 10 clearly show that MiZax3 is by far the most efficient of all the MiZax compounds that you used for comparisons in this Figure. This may be due to the fact that you did not choose the "best performers" from every group for comparison (which would be, for example, MZ7, MZ3-2, and MZ5-6, but definitely not MZ6, MZ3-1, and MZ5-2, which I believe you just randomly chose as the firstly-numbered compound from each group). The comparison shown in Figure 10 makes sense if your intention was to show that the newly synthesized MiZax compounds do not perform better than the previously available MZs (especially MiZax3). If that was not your intention, you may safely remove Figure 10 (and the entire section 3.4) from the manuscript, but an even better idea could be to keep it, adding an explanation that it shows how MiZax3 is more useful to control the Striga infestation than any of the newly synthesized compounds.

·         Discussion:

o    As previously said, your entire Discussion section needs to be completely re-written, pointing out the fact that, although certain newly synthesized MiZax compounds did outperform MiZax3 and/or MiZax5 in terms of isolated plant growth promotion traits or reduction in the amount of individual SL compounds in rice root exudates, none of the newly synthesized compounds showed consistently better performance characteristics compared to MiZax3 or MiZax5. With the 14 newly synthesized compounds and the 2 old ones (MiZax3 and MiZax5), you now dispose of a sizeable collection of 16 synthetic compounds that can be used to enhance the growth of rice plants and reduce their infestation by Striga. Ask yourselves: if you were rice farmers and could choose between these 16 compounds (including the original MiZax3 and MiZax5), which is the single one that you would want to apply to your rice plantation hoping for the best outcome in terms of economical profit? Based on the sizeable amount of results that you obtained from your research, you should be able to pick one.

o    It is still okay if the "best performer" is none of the newly synthesized compounds, but one of the "original" ones (MZ3 or MZ5). But in that case, you should acknowledge that your research revealed that, although trying many structural modifications to the original MZ3 and MZ5 compounds, you did not manage to synthesize one with overall better characteristics, which still means that your originally synthesized compounds are very good and may be used for controlling the infestation of rice plantations by Striga.

o    If the results that you presented here are still not sufficient to decide which MiZax is the "best performer", you might consider performing a principal component analysis (PCA) of your results to get a graphical answer to this question. This could be an interesting, and useful, addition to your Results section.

o    When you get a decisive answer to this question (or at least a couple of most likely answers), you should center your entire Discussion (and Conclusions as well) around this observation. You may discuss the differences in performance between the individual MiZax compounds and align them with the corresponding differences in chemical structure, but please avoid the vague statements, or statements devoid of meaning, such as:

§  "substituting the ketone by imine, alcohol, and ether groups to generate MZ3 derivatives led to enhanced root biomass, such in the case of MZ3-2" (line 505-506) (this commentary is essentially wrong because only one of the mentioned structural modifications was applied to MZ3-2)

§  "suggesting that this modification could improve the bioactivities" (line 551-552) (in some of the mentioned compounds this modification did improve the bioactivity, but in others it actually weakened it, so this statement is insufficiently supported by your results)

§  the Discussion is abundant with similar vague, insufficiently specific statements, or statements insufficiently supported by specific results. A good Discussion should rely on much more specific statements, which are tightly connected to specific results.

§  performing a PCA analysis might help you identify the connections between specific structural modifications and specific biological effects such as plant growth promotion, or reduction of Striga germination. For this reason, performing PCA might help you discuss your results, although this is, of course, not guaranteed.

o    Identifying a "single best performer" among the MiZax compounds should also help you write a more significant Conclusions section.

·         Author Contribution Statement: Please write in line with the recommendations given by CRediT Taxonomy, as presented in: https://credit.niso.org/

Author Response

Response to Reviewer-1

biomolecules-2493390

Dear Editor,

Thank you for considering our manuscript entitled “New series of zaxinone mimics (MiZax) for fundamental and applied research” by Jamil et al. for publication in the biomolecules-Special issue on Plant Growth Regulators for Stress Management in Plants.

We have addressed and incorporated all the suggestions raised by the reviewer and hope that our manuscript will find acceptance now.

Please find our response to the reviewers on separate pages.

With best regards

Salim Al-Babili

Comments and Suggestions for Authors

Dear Authors,

I have reviewed your manuscript "New series of zaxinone mimics (MiZax) for fundamental and applied research", submitted for publication in Biomolecules.

In your paper you presented a comprehensive research on the effects of novel MiZax compounds on rice plant growth, the inhibition of strigolactone biosynthesis and the infestation of rice plants with Striga hermonthica. Unfortunately, although some of the 14 newly synthesized compounds displayed a moderately stronger effect on individual parameters of rice plant growth or Striga infestation, none of them seems to overall outperform the effects of the highly efficient MiZax3 and MiZax5. In other words, if I was a rice farmer and had to pick a MiZax compound to treat my plants with, based on your results I would still pick MiZax3 or MiZax5 over any of the newly synthesized compounds you presented in your new paper.

The fact that you did not prove a consistently superior performance of any of the 14 newly synthesized compounds compared to MiZax3 and MiZax5, does not discredit your research in my opinion - your work proves that subtle structural modifications to the MiZax compounds do not importantly improve their agronomical performance, and might suggest, that expectations from novel compounds obtained through further structural modifications of MiZax should be lowered. However, although this conclusion is obvious to the reader after seeing the results of your current research, you did not point it out. Although you compared the effects of the novel MiZax with the previously available MiZax3 and MiZax5 throughout the Results section, your Discussion and Conclusions are focused on praising the novel MiZax compounds and their biological effects, comparing them mostly to untreated (control) plants, as if the (mostly better-performing) MiZax3 and MiZax5 were not already available. The Discussion section mostly relies on vague statements about the effects of structural modifications of MiZax on enhancing their performance, but the particular effects of structural modifications are never discussed - and neither are the novel compounds ever compared to MiZax3 and MiZax5 in terms of overall performance.

My opinion on your paper is that you should completely refocus the tone of your discussion, and offer a more critical comparison between the novel MiZax compounds and their "parent compounds" MiZax3 and MiZax5. I would urge you, when revising your paper, please do not keep the deleted text in tracked changes, but instead only highlight the new portions of text which have been added or revised. For more detailed remarks on your paper, please follow my comments as given below:

  • General language concerns:

o    please put Striga in italic letters throughout the manuscript text

Response:

We have corrected throughout the manuscript as suggested.

o    please use the word "infestation" instead of "infection" throughout the manuscript text. "Infection" is used for microorganisms, whereas "infestation" is more appropriate for macroscopic organisms such as parasitic pests or weeds.

Response:

We have corrected throughout the manuscript as suggested.

  • Introduction:

o    line 32: The term "food insecurity" already encompasses the risks of human starvation and malnutrition. Please revise.

Response:

Revised.

o    line 35: sub-Saharan (a letter A is missing)

Response:

Corrected.

o    line 55: "The evolutionarily-conserved SL biosynthetic pathway..." - please start a new paragraph with this sentence. The description of the biosynthetic pathway is very complex and should have a whole paragraph dedicated to it to help the reader maintain focus.

Response:

Corrected.

o    line 58-59 and elsewhere: Please check the text for a possible presence of double-spaces.

Response:

Corrected.

o    line 62: "...and is catalyzed by..." - please replace "is" with "are"

Response:

Corrected.

o    line 69: please delete "Scheme of" from the Figure caption

Response:

Deleted.

o    Figure 2 and line 76: Please also provide the structure of zaxinone in Figure 2

Response:

Corrected.

o    line 78: the original work in which zaxinone was discovered (Wang et al. 2019 published in Nature Communications: https://doi.org/10.1038/s41467-019-08461-1), should be provided here, instead of ref.35.

Response:

Reference is added.

o    lines 86-94: an explanation regarding the nomenclature of MiZax6, 7, and 8 vs. MiZax3-1, 3-2, 3-3, and 3-4 should be provided here. All these seven compounds have been derived from MiZax3, but only four of them (MiZax3-1 through 3-4) have been named after MiZax3, whereas MiZax6, 7, and 8 have received "names of their own". In which aspect do these two groups of MiZax3 derivatives differ from each other, and why were different nomenclature rules applied to them? A brief explanation should be provided within the Introduction section. Alternatively, if there is no good explanation but these names have been given ad hoc, I would suggest to rename MiZax6, 7, and 8 to MiZax3-6, 3-7, and 3-8, respectively, and to revise, accordingly, the entire manuscript.

Response:

Corrected.

o    also lines 86-94: a clear statement of your research goal is missing from the end of Introduction. Why did you synthesize novel MiZax compounds and test their performance in terms of rice plant growth promotion, strigolactone synthesis, and infestation with Striga? Highly efficient MiZax3 and MiZax5 have already been synthesized and tested for these effects, with very satisfying results. Why would you need novel compounds and test them? My guess is that you wanted to investigate whether, through a diverse set of structural modifications of the highly efficient compounds MiZax3 and MiZax5, you could obtain novel compounds with even greater biological activity in terms of the above-mentioned desired effects. This research goal should be more clearly stated at the end of your Introduction, and the Discussion and Conclusions to your work should be formulated so as to position your obtained results in relation to whether or not this research goal has been met, to what extent, and how; or why not.

Response:

We thank the reviewer and we have added information as suggested.

  • Materials & Methods:

o    line 103: Please provide the affiliation of the Late Prof. Babiker.

Response:

Added.

o    line 106: includeD

Response:

Corrected.

o    line 110: The synthesis scheme and the physico-chemical properties of the 14 newly synthesized MiZax compounds are not provided in Figure 2. Figure 2 only shows their structural formulae. By contrast, the synthesis scheme and the physico-chemical properties of the MiZax compounds are presented in the continuation of the M&M section, which is very useful, detailed and informative in terms of content, but on the other hand it is very poorly structurally defined, in terms of what subsection it belongs to, and whether it consists of text, of figures, or of combination of text and figures. This part of the manuscript (pages 4-9 and start of page 10) calls for clearer organizations into sections, subsections, sub-subsections and/or Figures - or, it can be entirely transferred to the Supplementary material. In any case, it cannot stay the way it is organized right now, because its relation to the manuscript structure is completely unclear.

Response:

We have modified as suggested.

o    line 112: What aldehyde?

Response:

Corrected.

o    line 201 and 209: "2N-Na2CO3" - what does the dash ("-") between "2N" and "Na2CO3" stand for? Did you mean "2N Na2CO3"? Please double-check, and revise if necessary. Same comment applies to the chemical reaction schemes in line 214.

Response:

Corrected.

o    page 7, other comments: please fully spell out THF and AcOEt at first mention, as well as all other abbreviations for compounds where standard IUPAC names are not used.

Response:

Added.

o    line 279: please rename the title of section 2.3 to "Rice Cultivation Conditions". "Details of Experiments" really does not mean anything.

Response:

Corrected.

o    line 289-290: Seedlings were treated with MiZax how? Transferred to fresh nutrient media containing MiZax, or did they have the MiZax solution poured on top of them? Please clearly describe the details of the experimental procedure within the M&M text.

Response:

Corrected.

o    line 293-294: please define the "low phosphate conditions"

Response:

Added information as Table S1.

o    line 294-295: If I understand correctly, the SLs were not "collected from root exudates", but instead, the root exudates were collected using a C18 column, and the SLs were isolated from the root exudates, as described in Section 2.5. Please be precise and clear in your statements.

Response:

We have made it clear.

o    line 295: please add "as described further" to the end of the sentence (after "Striga seeds"), to prepare the reader that the procedure is actually about to be explained in the immediate follow-up.

Response:

We have made it clear.

o    line 307: please add the word "rice" (RICE seedlings)

Response:

Added.

o    line 308: What is the bio-chamber now? I do not recall you introducing a bio-chamber to the reader.

Response:

We have made it clear.

o    Section 2.6: you used capital L for "liter" in this section, whereas you used lowercase "l" everywhere else in the manuscript. Please stick uniformly to either capital L or lowercase "l" (I must say that I personally prefer the capital L) throughout the entire manuscript.

Response:

Corrected.

o    please add an additional subsection to Materials & Methods, to describe the software used for statistical analysis.

Response:

Added.

  • Results:

o    General remark about the Results section: It is very tiring for the reader to read three sets of very similar results one after another. The Authors should consider reorganizing the entire Results section to pull together all the results for all the 14 novel MiZax compounds, either in single graphs, or at least in separate graphs but presented immediately one after another, so that the reader would not jump back and forth between plant growth parameters, SL composition and the Striga germination parameters. Thus, the Figures 3, 6, and 8 could be combined into a single Figure (or they could remain as they currently are, but grouped together one after another). For Figures 4, 7, and 9, I suggest that the results referring to the SL composition in root exudates should be separated from the results referring to Striga germination (they should be placed in separate Figures, and also, all the results referring to the SL composition should be presented before all the results for Striga germination, for all the MiZax compounds).

Response:

We thank the reviewer for the valuable comment. We have made all changes as suggested.

o    Also, in their narration of the Results, the Authors should stick strictly to the order in which the results are presented in the Figures. Currently, in the paragraph 360-374, the Authors are jumping back and forth between narrating the results presented in the Figures 4 and 5.

Response:

We thank the reviewer for the valuable comment. We have made all changes as suggested.

o    The subfigure "A" in Figures 3 and 4 should be made a lot larger (at least as large as it is in the case of Figure 5) and more clearly spatially separated from the subfigures B, C, and D.

Response:

We have revised the figures.

o    Figure captions: the letters p (for confidence intervals) and n (for sample size) should be written in italic letters in all the figure captions throughout the manuscript.

Response:

Corrected.

o    Figures 3-9 (except for Figure 5): in the Figure captions, you state that the "asterisks denote significance determined through one-way ANOVA". However, each asterisk is applied to the entire group of results (for instance, in Figure 3B, the three asterisks mean that the differences within the group are statistically significant at the p < 0.005 level). I find that the Fisher's LSD (least significant differences) test for the determination of the statistical significance of the differences between each treatment and the blank control would be much more informative. Alternatively, instead of comparing each of the MiZax compounds with the blank control, you might want to compare it to the effect of the "parental" MiZax compound (for instance, MiZax3-1 with MiZax3), because this is what is really of greatest interest within your current research.

Response:

We have revised the figures.

o    Speaking of control, you have used terms "blank", "mock" and "control" even within the same Figure (for instance, in Figures 4, 7, and 9). Please pick just one of these synonyms ("blank", "mock", or "control") and use it consistently throughout your manuscript, including within all the Figures and Figure captions.

Response:

We have revised the figures.

o    line 394: MZ3-4, not "MZ-4"

Response:

Corrected.

o    in the Results section (for instance, lines 408, 411-412) you use the term "germinating activity" for "germination efficiency". Please revise, and thoroughly check throughout the entire manuscript text.

Response:

Corrected.

o    line 457: the maximum, 36% reduction in Striga germination was obtained for the original compound MZ5, from which the other MiZax5s were derived. Thus, the MZ5 derivatives showed a reduction in Striga germination of up to 34% (for MZ5-3).

Response:

We have revised as suggested.

o    line 459: not "the same bioactivity", but "similar bioactivity".

Response:

Corrected.

o    Section 3.4 and Figure 10: The results in Figure 10 clearly show that MiZax3 is by far the most efficient of all the MiZax compounds that you used for comparisons in this Figure. This may be due to the fact that you did not choose the "best performers" from every group for comparison (which would be, for example, MZ7, MZ3-2, and MZ5-6, but definitely not MZ6, MZ3-1, and MZ5-2, which I believe you just randomly chose as the firstly-numbered compound from each group). The comparison shown in Figure 10 makes sense if your intention was to show that the newly synthesized MiZax compounds do not perform better than the previously available MZs (especially MiZax3). If that was not your intention, you may safely remove Figure 10 (and the entire section 3.4) from the manuscript, but an even better idea could be to keep it, adding an explanation that it shows how MiZax3 is more useful to control the Striga infestation than any of the newly synthesized compounds.

Response:

Added.

  • Discussion:

o    As previously said, your entire Discussion section needs to be completely re-written, pointing out the fact that, although certain newly synthesized MiZax compounds did outperform MiZax3 and/or MiZax5 in terms of isolated plant growth promotion traits or reduction in the amount of individual SL compounds in rice root exudates, none of the newly synthesized compounds showed consistently better performance characteristics compared to MiZax3 or MiZax5. With the 14 newly synthesized compounds and the 2 old ones (MiZax3 and MiZax5), you now dispose of a sizeable collection of 16 synthetic compounds that can be used to enhance the growth of rice plants and reduce their infestation by Striga. Ask yourselves: if you were rice farmers and could choose between these 16 compounds (including the original MiZax3 and MiZax5), which is the single one that you would want to apply to your rice plantation hoping for the best outcome in terms of economical profit? Based on the sizeable amount of results that you obtained from your research, you should be able to pick one.

Response:

We have added this aspect in the discussion.

o    It is still okay if the "best performer" is none of the newly synthesized compounds, but one of the "original" ones (MZ3 or MZ5). But in that case, you should acknowledge that your research revealed that, although trying many structural modifications to the original MZ3 and MZ5 compounds, you did not manage to synthesize one with overall better characteristics, which still means that your originally synthesized compounds are very good and may be used for controlling the infestation of rice plantations by Striga.

Response:

We have added this aspect in the discussion.

o    If the results that you presented here are still not sufficient to decide which MiZax is the "best performer", you might consider performing a principal component analysis (PCA) of your results to get a graphical answer to this question. This could be an interesting, and useful, addition to your Results section.

Response:

We thank the reviewer for this valuable comment. The principal component analysis (PCA) of the results could be an interesting and useful addition but we cannot do the PCA analysis from the present findings because we did not screen all new compounds at the same time.

o    When you get a decisive answer to this question (or at least a couple of most likely answers), you should center your entire Discussion (and Conclusions as well) around this observation. You may discuss the differences in performance between the individual MiZax compounds and align them with the corresponding differences in chemical structure, but please avoid the vague statements, or statements devoid of meaning, such as:

  • "substituting the ketone by imine, alcohol, and ether groups to generate MZ3 derivatives led to enhanced root biomass, such in the case of MZ3-2" (line 505-506) (this commentary is essentially wrong because only one of the mentioned structural modifications was applied to MZ3-2)

Response:

We have revised the discussion.

  • "suggesting that this modification could improve the bioactivities" (line 551-552) (in some of the mentioned compounds this modification did improve the bioactivity, but in others it actually weakened it, so this statement is insufficiently supported by your results)

Response:

Corrected.

  • performing a PCA analysis might help you identify the connections between specific structural modifications and specific biological effects such as plant growth promotion, or reduction of Strigagermination. For this reason, performing PCA might help you discuss your results, although this is, of course, not guaranteed.

Response:

We thank the reviewer for this valuable comment. The principal component analysis (PCA) of the results could be an interesting and useful addition but we cannot do the PCA analysis from the present findings because we did not screen all new compounds at the same time.

o    Identifying a "single best performer" among the MiZax compounds should also help you write a more significant Conclusions section.

Response:

We have revised as suggested.

  • Author Contribution Statement: Please write in line with the recommendations given by CRediT Taxonomy, as presented in: https://credit.niso.org/

Response:

Corrected.

Reviewer 2 Report

The authors present the evaluation of a novel series of Zaxinone mimics and evaluate their performance in terms of plant development and ability to control the germination of striga seeds.
The article is well written and each part contains appropriate literature references.

My major comments to work are:

-          It is clear a general positive effect on the development of the root apparatus for the MiZaxs.

Whereas the interpretation of the relation among the alteration on the composition of the SL exuded, the % of striga germination and the effective reduction on striga emergence is more complex. However I wonder if there is any effect due to the particular way by which the MiZaxs are administered (root treatment vs. foliar application, for instance). I believe a comment would be valuable in the text, if appropriate.

-          Second comments, is about the chemical modification of the MiZaxs. I’m not a biochemistry but I wonder if chemical modifications may have any chance to have a toxicological relevance in terms of application in agriculture and for organisms in general. Perhaps a general comment concerning the use of such molecules in open fields would be interesting.

My minor comments:

-          The concentration of MiZaxs at 1 uM in the different assay, I assume it has been previously defined in precedent works. It should be clarified.

-          Line 288 : clarify “modified Hoagland”

-          Line 289: “Seedling were treated”, please be more specific indicating if root application has been performed

-          Line 309: same comment as before

-          Line 336: is 5uM referred to MiZaxs or P?

-          Line 359: the level of significancy  ** and *** are the same, please control also all the other figures in the text.

-          Line 385: gene expression analysis is missing in materials and methods, it has to be included.

Author Response

Response to Reviewer-2

biomolecules-2493390

Dear Editor,

Thank you for considering our manuscript entitled “New series of zaxinone mimics (MiZax) for fundamental and applied research” by Jamil et al. for publication in the biomolecules-Special issue on Plant Growth Regulators for Stress Management in Plants.

We have addressed and incorporated all the suggestions raised by the reviewer and hope that our manuscript will find acceptance now.

Please find our response to the reviewers on separate pages.

With best regards

Salim Al-Babili

Comments and Suggestions for Authors

The authors present the evaluation of a novel series of Zaxinone mimics and evaluate their performance in terms of plant development and ability to control the germination of striga seeds.
The article is well written and each part contains appropriate literature references.

My major comments to work are:

-          It is clear a general positive effect on the development of the root apparatus for the MiZaxs. Whereas the interpretation of the relation among the alteration on the composition of the SL exuded, the % of striga germination and the effective reduction on striga emergence is more complex. However, I wonder if there is any effect due to the particular way by which the MiZaxs are administered (root treatment vs. foliar application, for instance). I believe a comment would be valuable in the text, if appropriate.

Response:

We thank the reviewer for the valuable comment. We have added a comment in the discussion section.

-          Second comments, is about the chemical modification of the MiZaxs. I’m not a biochemistry but I wonder if chemical modifications may have any chance to have a toxicological relevance in terms of application in agriculture and for organisms in general. Perhaps a general comment concerning the use of such molecules in open fields would be interesting.

Response:

We thank the reviewer for the comment. We have added a comment in the discussion section.

My minor comments:

-          The concentration of MiZaxs at 1 uM in the different assay, I assume it has been previously defined in precedent works. It should be clarified.

Response:

We have clarified as suggested.

-          Line 288 : clarify “modified Hoagland”

Response:

We thank the reviewer for the comment. We have added detail as a supplementary Table.

-          Line 289: “Seedling were treated”, please be more specific indicating if root application has been performed

Response:

We have specified in the respective method section.

-          Line 309: same comment as before

Response:

We have specified in the respective method section.

-          Line 336: is 5uM referred to MiZaxs or P?

Response:

We have specified in the respective method section.

-          Line 359: the level of significancy  ** and *** are the same, please control also all the other figures in the text.

Response:

We have made correction as suggested.

-          Line 385: gene expression analysis is missing in materials and methods, it has to be included.

Response:

We have added gene expression analysis in the method section.

Submission Date

23 June 2023

Date of this review

11 Jul 2023 19:14:00

Round 2

Reviewer 1 Report

Dear Authors,

I have been asked for a second round of review of your manuscript "New series of zaxinone mimics (MiZax) for fundamental and applied research", submitted for publication in Biomolecules.

Reading the revised version, I was pleased to see that you have successfully implemented most of my recommendations for improvement from the first round of review. Your manuscript has visibly improved, and I will recommend it for publication in Biomolecules after a final round of corrections that remain to be done so that your paper can make the maximum scientific impact. Please follow my recommendations as given below:

·         Abstract - A concluding sentence should be added to the Abstract, hinting at the outcome of your research with respect to your research goal (i.e., that the application of structural modifications to MiZax3 and MiZax5 did not additionally improve their overall performance and that these two compounds - especially MiZax3 - remain likely the most efficient zaxinone mimics for controlling Striga infestation) - something like your observations given in lines 442-443 in the Discussion or in 476-478 in the Conclusions section. This should be clearly stated also within the Abstract, so that the reader can find this important information just looking at the Abstract, without needing to scroll down to the Conclusions of your article.

·         Introduction - The Introduction section is now flawless.

·         Materials & Methods:

o    line 109: please replace "has shown" with "is shown" and "physio-chemical" with "physico-chemical"

o    line 112: please replace "have been shown" with "are shown"

o    line 112: Also, Table S1 should be cited here again, and not Table S2.

o    regarding Supplementary Tables, please correct their titles and call them "Table S1", "Table S2", "Table S3", instead of "Table 1", "Table 2", "Table 3".

o    Figure 3: please add very brief explanations under each arrow, as you did in Figure 4

o    lines 119 and 139: There is an open bracket before "Wakosil®", which is never closed thereafter.

o    Figures 4A and 4B are clearly separate Figures. Please renumber them as Figures 4 and 5, and renumber all the successive Figures accordingly, including the reference to the Figures within the manuscript text.

o    line 164-165: please revise into: "each of the three compounds synthesized in the first step, as well as MZ5 (each at 1.0 mM)"

o    line 183: please delete "and a 12 h photoperiod", since you literally just already said it previously, in the same sentence

o    line 218: please delete the space between "acetonitrile:" and "water"

o    line 253: please put "-ΔΔCt" into superscript.

o    line 258: If you stick to your previous choice not to show the results of the LSD test in your figures, but only the existence of significant differences within the groups of results according to ANOVA, then you should delete any mention of the LSD test within your Materials & Methods.

·         Results:

o    line 264: I believe it is a methoxy- and not a methyl-group

o    line 274: At the beginning of this paragraph, you should briefly summarize which structural modifications to MZ3 have been made to obtain MZ3-6, MZ3-7, and MZ3-8, similarly as you did for MZ3-1, MZ3-2, MZ3-3, and MZ3-4 in the previous paragraph.

o    line 283-284: "one or two phenyl-groups" - or also all three of them, as in the case of MZ5-8.

o    line 315: please correct "MZ3-5, -6, and -7" into "MZ3-6, -7, and -8"

o    line 317: I insist that you should add here, regarding oxo-CL, that the application of MZ3-8 interestingly increased the content of oxo-CL by more than 100% (Figure 7H) (or 8H if you renumber them). This is really an extraordinary result.

o    line 346-347: please revise to: "to ultimately result in the reduction of germination efficiency of Striga seeds, due to the altered composition of rice root exudates"

o    line 358: please correct to: "the least reduction in the Striga germination efficiency"

o    line 379: please correct to: "MZ5-2 and MZ3-6"

·         Discussion:

o    line 408: please either correct "SLs" to "SL", or, alternatively, put the whole sentence into plural.

o    line 433-434: I was very surprised to read this idea. A potential for application as a suicidal germination agent? You mean, by farmers who want to destroy their own crops? Please either explain or revise.

o    line 435: please correct to: "were not efficient in reducing the stimulation of Striga germination by rice root exudates"

·         Conclusions:

o    line 471: if you are mentioning 4-deoxyorobanchol here, then you should definitely also mention oxo-carlactone, since it was increased by more than 100% by the application of MZ3-8.

o    line 477: especially MiZax3 and it definitely deserves to be pointed out here, as well as at the end of the Abstract

·         Minor text corrections:

o    there are possible remaining double-spaces at multiple places throughout the manuscript (lines 60, 119, possibly elsewhere). Please double-check, and correct if necessary.

o    line 63: "are catalyzed" - this is a correction (from "is catalyzed") that I have requested during the first round of review. However, I now realize that "is catalyzed" actually makes more sense, and it always did. I want to apologize to the Authors for this mistake, and would like to ask them to revert the original version of this sentence.

Kind regards,

Reviewer 1

Author Response

Response to Reviewer-1-R2

biomolecules-2493390

Dear Editor,

Thank you for considering our manuscript entitled “New series of zaxinone mimics (MiZax) for fundamental and applied research” by Jamil et al. for publication in the biomolecules-Special issue on Plant Growth Regulators for Stress Management in Plants.

We have addressed and incorporated all the suggestions raised by the reviewer-1 again and hope that our manuscript will find acceptance now.

Please find our response to the reviewers on separate pages.

With best regards

Salim Al-Babili

Response to Reviewer -1

I have been asked for a second round of review of your manuscript "New series of zaxinone mimics (MiZax) for fundamental and applied research", submitted for publication in Biomolecules.

Reading the revised version, I was pleased to see that you have successfully implemented most of my recommendations for improvement from the first round of review. Your manuscript has visibly improved, and I will recommend it for publication in Biomolecules after a final round of corrections that remain to be done so that your paper can make the maximum scientific impact. Please follow my recommendations as given below:

  • AbstractA concluding sentence should be added to the Abstract, hinting at the outcome of your research with respect to your research goal (i.e., that the application of structural modifications to MiZax3 and MiZax5 did not additionally improve their overall performance and that these two compounds - especially MiZax3 - remain likely the most efficient zaxinone mimics for controlling Striga infestation) - something like your observations given in lines 442-443 in the Discussion or in 476-478 in the Conclusions section. This should be clearly stated also within the Abstract, so that the reader can find this important information just looking at the Abstract, without needing to scroll down to the Conclusions of your article.

Response:

Added.

  • Introduction- The Introduction section is now flawless.

  • Materials & Methods:

o    line 109: please replace "has shown" with "is shown" and "physio-chemical" with "physico-chemical"

Response:

Corrected.

o    line 112: please replace "have been shown" with "are shown"

Response:

Corrected.

o    line 112: Also, Table S1 should be cited here again, and not Table S2.

Response:

Corrected.

o    regarding Supplementary Tables, please correct their titles and call them "Table S1", "Table S2", "Table S3", instead of "Table 1", "Table 2", "Table 3".

Response:

Corrected.

o    Figure 3: please add very brief explanations under each arrow, as you did in Figure 4

Response:

Added.

o    lines 119 and 139: There is an open bracket before "Wakosil®", which is never closed thereafter.

Response:

Corrected.

o    Figures 4A and 4B are clearly separate Figures. Please renumber them as Figures 4 and 5, and renumber all the successive Figures accordingly, including the reference to the Figures within the manuscript text.

Response:

Corrected.

o    line 164-165: please revise into: "each of the three compounds synthesized in the first step, as well as MZ5 (each at 1.0 mM)"

Response:

Corrected.

o    line 183: please delete "and a 12 h photoperiod", since you literally just already said it previously, in the same sentence

Response:

Corrected.

o    line 218: please delete the space between "acetonitrile:" and "water"

Response:

Corrected.

o    line 253: please put "-ΔΔCt" into superscript.

Response:

Corrected.

o    line 258: If you stick to your previous choice not to show the results of the LSD test in your figures, but only the existence of significant differences within the groups of results according to ANOVA, then you should delete any mention of the LSD test within your Materials & Methods.

Response:

Corrected.

  • Results:

o    line 264: I believe it is a methoxy- and not a methyl-group

Response:.

Corrected.

o    line 274: At the beginning of this paragraph, you should briefly summarize which structural modifications to MZ3 have been made to obtain MZ3-6, MZ3-7, and MZ3-8, similarly as you did for MZ3-1, MZ3-2, MZ3-3, and MZ3-4 in the previous paragraph.

Response:

Added.

o    line 283-284: "one or two phenyl-groups" - or also all three of them, as in the case of MZ5-8.

Response:

Corrected.

o    line 315: please correct "MZ3-5, -6, and -7" into "MZ3-6, -7, and -8"

Response:

Corrected.

o    line 317: I insist that you should add here, regarding oxo-CL, that the application of MZ3-8 interestingly increased the content of oxo-CL by more than 100% (Figure 7H) (or 8H if you renumber them). This is really an extraordinary result.

Response:

Added.

o    line 346-347: please revise to: "to ultimately result in the reduction of germination efficiency of Striga seeds, due to the altered composition of rice root exudates"

Response:

Revised.

o    line 358: please correct to: "the least reduction in the Striga germination efficiency"

Response:

Corrected.

o    line 379: please correct to: "MZ5-2 and MZ3-6"

 Response:

Corrected.

  • Discussion:

o    line 408: please either correct "SLs" to "SL", or, alternatively, put the whole sentence into plural.

Response:

Corrected.

o    line 433-434: I was very surprised to read this idea. A potential for application as a suicidal germination agent? You mean, by farmers who want to destroy their own crops? Please either explain or revise.

Response:

Corrected.

o    line 435: please correct to: "were not efficient in reducing the stimulation of Striga germination by rice root exudates"

 Response:

Corrected.

  • Conclusions:

o    line 471: if you are mentioning 4-deoxyorobanchol here, then you should definitely also mention oxo-carlactone, since it was increased by more than 100% by the application of MZ3-8.

Response:

Mentioned.

o    line 477: especially MiZax3 and it definitely deserves to be pointed out here, as well as at the end of the Abstract

 Response:

Mentioned.

  • Minor text corrections:

o    there are possible remaining double-spaces at multiple places throughout the manuscript (lines 60, 119, possibly elsewhere). Please double-check, and correct if necessary.

 Response:

Corrected.

o    line 63: "are catalyzed" - this is a correction (from "is catalyzed") that I have requested during the first round of review. However, I now realize that "is catalyzed" actually makes more sense, and it always did. I want to apologize to the Authors for this mistake, and would like to ask them to revert the original version of this sentence.

Response:

Corrected.
